# Determinants of non-Hodgkin's lymphoma at Felegehiwot specialized hospital, North West Ethiopia: A case-control study

Dessalegn Chekol[1☯], Melkamu Bedimo[2☯], Yihun Mulugeta[2☯], Getasew Mulat Bantie[3☯]*

1 Department of Nursing, Felegehiwot Comprehensive Specialized Hospital, Bahir Dar, Ethiopia,
2 Department of Biostatistics and Epidemiology, Bahir Dar University, Bahir Dar, Ethiopia, 3 Department of Public Health, Faculty of Community Health, Alkan Health science, Business and Technology College, Bahir Dar, Ethiopia

☯ These authors contributed equally to this work.
* getasewmulat@gmail.com

**Data Availability Statement:** All the data can access from the manuscript.

**Funding:** The author(s) received no specific funding for this work.

## Abstract

### Background

The global burden of cancer continues to increase largely because of the aging and growth of the world population alongside an increasing adoption of cancer-causing behaviors. Hence, the purpose of this study was to identify determinants of Non-Hodgkin lymphoma cancer among individuals who diagnosed at the Felegehiwot specialized hospital, North West Ethiopia, 2019.

### Methods

An institution-based unmatched case-control study was conducted at the Felegehiwot Specialized hospital from December 2018 up to June 2019. The sample size calculated using the two-population proportion formula. The final sample size was 486, (162 cases and 324 controls). The simple random sampling method was employed to catch up with the estimated samples. The collected data entered into the Epi-data version 3.1 software and analyzed using SPSS version 21 software. Descriptive statistics computed. Simple logistic analysis was run (at 95% CI and p-value < 0.05) to identify the determinants of non-Hodgkin's lymphoma.

### Result

A total of 486 patients participated. Nearly one-third of the cases and controls were in the age group of 46–60 years. About 90% of cases and 91% of controls were orthodox Christian. Monthly income of ≤28 dollars (AOR = 2.73, 95%CI: 1.8, 4.2), male sex (AOR = 1.8, 95%CI: 1.2, 2.8), ever had chemical exposure, (AOR = 11.9, 95%CI: 7.6, 18.8), no regular physical exercise (AOR = 15.5, 95%CI: 5.7, 42.3), and having hypertension [AOR = 0.03; 95%CI:0.005, 0.2), lung disease (AOR = 0.2; 95%CI: 0.06, 0.7), and chronic kidney and cardiac diseases (AOR = 0.06; 95%CI: 0.01, 0.2) were the determinants of non-Hodgkin's lymphoma.

**Competing interests:** The authors have declared that no competing interests exist.

## Conclusions

The findings in this study suggest that having a low monthly income, being male sex, ever had chemical exposure, not engaged in regular physical exercise, and being diabetic were the determinants of non-Hodgkin's lymphoma.

## Background

In Africa, approximately 300, 000 cases of non-Hodgkin lymphoma (NHL) occur each year. The infections are among the top ten causes of cancer in this continent region [1]. In Ethiopia, studies showed that there are more than 150,000 cancer cases per year [2]. Reports indicated that 540 cancer cases from September 2014 to August 2015 seen at the University of Gondar Hospital, North-West Ethiopia, of whom 93 were lymphomas [2]. The estimated number of new cases in Ethiopia in 2018 is 67,573 (both sexes); Of whom 3,470 were NHL [3].

Lifestyle-related risk factors such as body weight, physical activity, diet, and tobacco use play a vital role in developing NHL [4]. Thus, identifying the risk factors for NHL may improve our understanding of the disease and is of very crucial for policymakers and other concerned stack holders at national as well as regional levels to design evidence-based intervention strategies to give emphasis and tackle the determinants of the NHL.

## Methods

### Study design, setting, and period

An institution-based unmatched case-control study was conducted at the Felegehiwot specialized hospital from November 2018 up to June 2019. The hospital found in Bahir Dar city, Amhara region, 564 km apart from Addis Ababa, the capital city of Ethiopia. It is the only public hospital in the Bahir Dar city, offering services to the dwellers of the region and the Amhara regional state. The hospital serves about five million people of the Amhara regional state communities. The hospital comprises of outpatient, inpatient, emergency, delivery, laboratory, ART, psychiatry, pharmacy, x-ray, physiotherapy, radiology, and oncology department. The oncology department of this hospital established in 2017. Since then, 169 patients attending at the oncology department registered in the logbook.

### Sample size determination

The sample size was determined using Epi- Info version 7 software, with the model of stat calc Fleiss W/CC. The identified determinants of NHL (reviewed from previously published articles) were cigarette smoking, coffee drinking, history of cancer, exposure to pesticides, and lifestyle [5–8]. Then, considering the result of these determinants, the predictor that has the largest estimated sample size was taken. Other common assumptions of 1:2 ratio of cases to controls, 80% power, adjusted odds ratio at 95% confidence interval, and a 5% margin of error used. Then, the final estimated samples were 486 (Table 1).

### Sampling procedure

**Cases.** Were NHL patients who diagnosed and confirmed histo-pathologically (162 out of 169). Then, the recruited patient medical record reviewed. Finally, histo-pathologically confirmed NHL patients interviewed during the study period.

**Table 1. Determinants for the sample size estimation of non-Hodgkin's lymphoma at Felegehiwot specialized hospital, northwest Ethiopia, 2019.**

| Variable | AOR | Proportions of the exposed cases | Proportions of the exposed controls | Required samples | | Total Samples | References |
|---|---|---|---|---|---|---|---|
| | | | | Cases | Controls | | |
| **Cigarette Smoking** | 2.4 | 50 | 29.4 | 65 | 129 | 194 | Fabbro-Peray, et al., 2001 |
| **Coffee drinking** | 2.9 | 94.1 | 84.6 | 141 | 281 | 422 | |
| **History of cancer** | 2.6 | 34.1 | 16.6 | 67 | 134 | 201 | Hardell, L. et al., 1999 |
| **Exposure to pesticides** | 3.7 | 10.3 | 3 | 115 | 229 | 334 | Balasubramaniam, G., et al., 2013 |
| **Lifestyle** | 2.79 | 94.5 | 86 | 162 | 324 | 486 | Cerhan, J.R., et al., 1997 |

**Controls.** Chronically ill medical patients who had non-cancerous pathology result and follow-up in the medical department were the controls. Two controls for each case took by a systematic sampling technique from the list of the service delivery logbook. Then, the recruited patient medical record reviewed; Finally, a chronically ill, non-cancerous, medical patient interviewed during the study period.

## Exclusion criteria

**Cases.** The seven histo-pathologically non-confirmed patients and not on follow up in the oncology department of the hospital were excluded.

**Controls.** Non-cancerous, chronically ill medical patients, who had not pathology result attached to their charts and not on follow-up in the medical department of the hospital excluded.

## Data collection tool and procedure

The data collected via primary and secondary data sources. The secondary sources of data retrieved from reviewing of the patient's medical records. Then, the primary data collected by using a pre-tested structured questionnaire adapted from a variety of literature [5–8] 'S1 File'. The questionnaire comprised of socio-demographic, feeding practice, and exposure to carcinogenic chemicals related characteristics. It translated into Amharic (the indigenous) language by the independent translator (Ph.D. in linguistics). Then, back to English to check for consistency. The principal investigator gave two days of rigorous training to two enumerators (BSc in clinical nurse) and one supervisor (MPH in Epidemiology). The enumerators conducted role-play before the actual data collection period. Finally, the data were collected using the Amharic version of the questionnaire. Each questionnaire examined daily for completeness and consistency by the supervisor and the principal investigator. Appropriate feedback gave to the enumerators.

## Data management and analysis

Data entry, cleaning, and coding were performed using Epi-data version 3.1 software and analysis done by SPSS version 20 software. Descriptive statistics were computed and presented using tables and texts. The bivariable regression model initially fitted to compute the crude odds ratio (COR), and variables with p-values less than 0.2 entered into the multivariable logistic regression model to control potential confounding effects in the model. The strength of associations between the determinants and the non-Hodgkin's lymphoma assessed using the adjusted odds ratio (AOR) with a 95% CI. Variables with p-values less than 0.05 in the multivariable analysis considered as the determinants of non-Hodgkin's lymphoma.

### Ethics approval and consent to participate

Ethical approval obtained from the Institutional Review Board of Bahir Dar University. Permission letter was obtained from the Amhara National, Regional state Health Bureau and Public Health Institute prior to the data collection period. We authors can assure that the Institutional Review Board of Bahir Dar University waived the need for parental consent for minority groups. Before collecting the data, for patients whose age 7–12 years written consent received from parents/guardians and assent of patients; and for 13–17 years old patients,' assents secured solely from them with parental/guardian permission. For older than 17 years patients, consent received solely from them. The names of the patients did not use to ensure anonymity and confidentiality. All information obtained from the patients was kept confidential.

## Results

### Socio-demographic characteristics of the respondents

A total of 486 (162 cases and 324 controls) study participants at followup in the Felegehiwot specialized hospital were interviewed and yielded a response rate of 100%.

A larger portion of cases (34%) and controls (35.2%) were in the age group of 46–60 years. About ninety percent of cases and controls were Orthodox Christian. The majority of the cases (84%) and controls (84.3%) were from the Amhara ethnic group. About two-thirds of cases and controls were unable to read and write. The majority of the cases and controls were married. More than three-fourths of the respondents come from a rural area, and more than one-fourth of the respondents were farmers (Table 2).

### Chemical exposure-related characteristics

About seventy-eight percent of the cases and twenty-two percent of the controls had previous exposure to carcinogenic chemicals. More than one-third of the respondents exposed to Herbicides. Half of the respondents exposed for more than fifteen years. About eight percent of the cases and five percent of the controls were cigarette smokers. Fifty-four percent of cases and sixty-two percent of controls smoked for more than ten years, respectively. Eighty-six percent of cases and sixty-seven percent of controls drunk. Eighty-eight percent of cases and eighty-three percent of controls drunk cultural alcohol. Eighty-six percent of cases and seventy-four percent of controls were coffee drunkest. Of which, three-fourth of them drunk coffee for more than thirty years (Table 3).

### Behavioral and feeding practice-related characteristics

More than two-thirds of the respondents had regular physical exercise, and more than half of the cases had a history of chronic illness. More than one-third of the cases (40%) and one-fourth of controls (26%) had lung disease (COPD). Two-third of the cases and about sixty percent of controls didn't know their HIV status. More than one-fourth of the cases and controls have started treatment on time (Table 4).

### Factors associated with non-Hodgkin's lymphoma

In the univariate logistic regression analysis, sex, residence, occupational status, monthly income, ever had chemical exposure, drinking alcohol, drinking coffee, had a regular physical exercise, type of chronic diseases, and HIV screening status were factors associated with non-Hodgkin's lymphoma at a 20% level of significance. In the multivariable logistic regression analysis, only sex, monthly income, ever had chemical exposure, had a regular physical

**Table 2. Sociodemographic characteristics of the respondents at Felegehiwot specialized hospital, northwest Ethiopia, 2019.**

| Variable | Category | Cases N (%) | Controls N (%) | $X^2$, P-Value |
|---|---|---|---|---|
| Age | <15 years | 10 (6.2) | 4 (1.2) | 16.691, 0.005 |
| | 15–30 years | 21 (13.0) | 59 (18.2) | |
| | 31–45 years | 33 (20.4) | 84 (25.9) | |
| | 46–60 years | 55 (33.9) | 114 (35.2) | |
| | 61–75 years | 36 (22.2) | 46 (14.2) | |
| | >75 years | 7 (4.3) | 17 (5.3) | |
| Sex | Male | 106 (65.4) | 177 (54.6) | 5.182, 0.023 |
| | Female | 56 (34.6) | 147 (45.4) | |
| Residence | Urban | 16 (9.9) | 79 (24.4) | 14.451, 0.0001 |
| | Rural | 146 (90.1) | 245 (75.6) | |
| Religion | Orthodox | 146 (90.1) | 295 (91.1) | 3.29, 0.193 |
| | Protestant | 5 (3.1) | 3 (0.9) | |
| | Muslim | 11 (6.8) | 26 (8.0) | |
| Ethnicity | Amhara | 136 (84.0) | 273 (84.3) | 0.008, 0.930 |
| | Non-Amhara | 26 (16.0) | 51 (15.7) | |
| Educational status | Unable to read and write | 106 (65.4) | 216 (66.6) | 2.207, 0.531 |
| | Elementary | 41 (25.3) | 67 (20.7) | |
| | Secondary | 9 (5.6) | 23 (7.1) | |
| | Diploma and above | 6 (3.7) | 18 (5.6) | |
| Occupation | Farmer | 70 (43.2) | 100 (30.9) | 11.255, 0.047 |
| | Housewife | 43 (26.5) | 91 (28.1) | |
| | Government employee | 6 (3.7) | 27 (8.3) | |
| | Military | 3 (1.9) | 3 (0.9) | |
| | Factory worker | 12 (7.4) | 25 (7.7) | |
| | Day laborer | 28 (17.3) | 78 (24.1) | |
| Marital Status | Married | 107 (66.0) | 197 (60.8) | 1.269, 0.260 |
| | Unmarried | 55(34.0) | 127 (39.2) | |
| Monthly Income | ≤28 USD dollar | 112 (69.1) | 141 (43.5) | 28.398, 0.0001 |
| | >28 dollar | 50 (30.9) | 183 (56.5) | |

USD: United States Dollar

exercise, and type of chronic disease were the determinants of non-Hodgkin's lymphoma at p = 0.05.

Accordingly, for those male participants, the odds of non-Hodgkin's lymphoma was about two (AOR = 1. 8, 95% CI:1.2, 2.8) fold higher compared with female participants. Similarly, those participants whose monthly income of equal or less than 28 dollars, the odds of non-Hodgkin's lymphoma were about three (AOR = 2. 73,95% CI:1.8, 4.2) times higher compared with income of beyond 28 dollars.

Those participants who had a history of chemical exposure, the odds of non-Hodgkin's lymphoma were twelve (AOR = 11. 98, 95%CI: 7.62, 18.85) times higher compared with those who hadn't a history of chemical exposure. Those participants who hadn't regular physical exercise, the odds of non-Hodgkin's lymphoma were about fifteen (AOR = 15. 5, 95% CI: 5.7, 42.3) times higher compared with those who had regular physical exercise. Moreover, participants who had hypertension (AOR = 0. 03, 95%CI: 0. 005, 0.18), chronic obstructive pulmonary disease (AOR = 0. 2, 95%CI: 0.06, 0.7), chronic kidney and cardiac diseases (AOR = 0. 06,

**Table 3. Chemical exposure-related characteristics of the respondents at Felegehiwot specialized hospital, northwest Ethiopia, 2019.**

| Variable | Category | Cases | Control | $X^2$, P-Value |
|---|---|---|---|---|
| | | N (%) | N (%) | |
| Ever had chemical exposure | No | 36 (22.2) | 251 (77.5) | 136.32, 0.001 |
| | Yes | 126 (77.8) | 73 (22.5) | |
| Smoking Cigarette | No | 149 (92.0) | 309 (95.4) | 2.29, 0.130 |
| | Yes | 13 (8.0) | 15 (4.6) | |
| Years of smoked | ≤ 10 years | 6 (46.2) | 5 (33.3) | 0.480, 0.488 |
| | > 10 years | 7 (53.8) | 10 (66.7) | |
| Alcohol drink | No | 22 (13.6) | 116 (35.8) | 26.231, 0.001 |
| | Yes | 140 (86.4) | 208 (64.2) | |
| Types of alcohol taken | Beer | 16 (11.4) | 36 (17.3) | 2.276, 0.131 |
| | Cultural alcohol | 124 (88.6) | 172 (82.7) | |
| Coffee drink | No | 22 (13.6) | 83 (25.6) | 9.239, 0.002 |
| | Yes | 140 (86.4) | 241 (74.4) | |
| Years of drinking coffee | < 20 years | 14 (10.0) | 32 (13.3) | 0.896, 0.344 |
| | ≥ 20 years | 126 (90.0) | 209 (86.7) | |

95%CI: 0.01,0.2) were less likely at risk of non-Hodgkin's lymphoma compared with those who had diabetes mellitus (Table 5).

## Discussion

### Monthly income

This study investigated the determinants of non-Hodgkin's lymphoma. It was recognized that monthly income as one of the crucial determinants of non-Hodgkin's lymphoma. Respondents who had monthly income of less than or equal to 28 dollars were about threefold more at risk to NHL compared to those who had greater than 28 dollars. A study done at Glostrup University Hospital, Denmark [9], supports the current study finding. The possible justification for this could be, little income enforces them to a low quality of living. As a result, it may expose to different forms of cancer like NHL. It is also could be justified, even if the study participants had health-seeking behavior, they become tied to not gain it due to the inability to cover the cost of the service. That might also increase the risk of non-Hodgkin's lymphoma.

### Sex

The odds of having an NHL in males were 2 times higher compared to females. This is in line with the study findings of Yale University, USA [10]. The reason for this risk difference between the two sexes could be males in our country are highly exposed to out-door work. And the majority of the respondents in our study were farmers, in which their work mainly confined to farm work. As a result, they may be more exposed to carcinogenic chemicals like herbicides and pesticides. This result supports the study findings of Yale University, USA [10], Sweden [8], and meta-analysis report [11].

### Ever had chemical exposure

Ever had a history of exposure for chemicals were found to be the root cause for the NHL. It found out in this study that; respondents who had chemical exposure were 12 times more likely at risk for NHL than those who had not. This result supports the findings of Yale

**Table 4. Behavioral and feeding practice-related characteristics of the respondents at Felegehiwot specialized hospital, northwest Ethiopia, 2019.**

| Variable | Category | Cases | Control | $X^2$, P-Value |
|---|---|---|---|---|
| | | N (%) | N (%) | |
| Regular physical exercise | Yes | 35 (21.6) | 120 (37.0) | 11.84, 0.001 |
| | No | 127 (78.4) | 204 (63.0) | |
| Had chronic diseases other than cancer | No | 75 (46.3) | _ | - - - - - - |
| | Yes | 87 (53.7) | 324 (100) | |
| Types of a chronic disease | Diabetes Mellitus | 25 (28.7) | 73 (22.5) | 17.27, 0.002 |
| | Liver disease | 4 (4.6) | 32 (9.9) | |
| | Hypertension | 3 (3.4) | 53 (16.4) | |
| | Lung disease (COPD) | 35 (40.2) | 82 (25.3) | |
| | Other[€] | 20 (23.0) | 84 (25.9) | |
| HIV status | No | 109 (67.3) | 189 (58.3) | 15.58, 0.001 |
| | Yes | 26 (16.0) | 30 (9.3) | |
| | Unknown | 27 (16.7) | 105 (32.4) | |
| Previous feeding practice | Meat | 17 (10.5) | 36 (11.1) | 5.92, 0.205 |
| | Vegetation | 9 (5.6) | 15 (4.6) | |
| | Non- vegetation | 2 (1.2) | 12 (3.7) | |
| | Milk | 7 (4.3) | 28 (8.6) | |
| | Other[€€] | 127 (78.4) | 233 (71.9) | |
| You got treatment for the chronic disease | No | 19 (11.7) | 32 (9.9) | 0.39, 0.530 |
| | Yes | 143 (88.3) | 292 (90.1) | |
| Outcomes of the treatment | Progressed | 112 (78.3) | 203 (79.5) | 4.35, 0.226 |
| | Deteriorated | 8 (5.6) | 20 (6.8) | |
| | Same | 22 (15.4) | 66 (22.5) | |
| | Not mentioned | 1 (0.6) | 3 (1.0) | |
| BMI of the patient | <18.5 kg/h$^2$ | 49 (30.2) | 227 (70.1) | - - - - - - |
| | 18.5–24.9 kg/h$^2$ | 113 (69.8) | 96 (29.6) | |
| | 25–29.9 kg/h$^2$ | 0 (0.0) | 1 (0.3) | |
| Where treated when the disease starts | Traditional medicine | 26 (16.0) | 32 (9.9) | 4.61, 0.1 |
| | Modern medicine | 78 (48.1) | 180 (55.6) | |
| | Holy water[Ω] | 58 (35.8) | 112 (34.6) | |

Other[€]: Chronic kidney disease and cardiac failure; other[€€]: nutritional intake of 'Injjera with watt' or Bread; holy water[Ω]: water blessed by a priest and used in religious ceremonies.

University, USA [10], Sweden [8], and meta-analysis report [11]. However, a study finding in the USA showed that respondents who exposed to chemicals protected from NHL [12].

Because the majority of chemicals are carcinogenic by their nature, and the current study showed four-fold riskier compare to the studies conducted in developed countries. The other reason could be having inadequate awareness and knowledge of the dangerous chemicals.

## Regular physical exercise

Not having regular physical exercise and the type of chronic illness, they had, were the determinants of the NHL. The odds of being caught by NHL from non-regular physical exercising respondents were about five times more likely at risk than those who did in the past. This finding is in line with the studies of California [13] and Canada, Ottawa [14]. There is a risk difference between the current study and the latter two studies. Because in Ethiopia, regular physical exercise is not habitual compared to California and the Canadian population.

**Table 5. Simple logistic regression on determinants of NHL at Felegehiwot specialized hospital, northwest Ethiopia, 2019.**

| Variable | Category | NHL | | COR (95% CI) | AOR (95% CI) | P-value |
|---|---|---|---|---|---|---|
| | | Cases | Controls | | | |
| **Sex** | Male | 106 | 177 | 1.6 (1.1, 2.3) | 1.8 (1.2, 2.8) | 0.004 |
| | Female | 56 | 147 | 1.00 | 1.00 | |
| **Residence** | Urban | 16 | 79 | 0.34 (0.2,0.6) | 0.55 (0.3, 1.1) | 0.061 |
| | Rural | 146 | 245 | 1.00 | 1.00 | |
| **Occupation** | Farmer | 70 | 100 | 1.00 | 1.00 | |
| | Housewife | 43 | 91 | 0.7 (0.4, 1.1) | 2.3 (0.6, 3.0) | 0.161 |
| | Government employee | 6 | 27 | 0.3 (0.1,0.8) | 0.93 (0.3, 2.7) | 0.891 |
| | Military | 3 | 3 | 1.4 (0.3, 7.3) | 1.69 (0.3, 9.7) | 0.556 |
| | Factory workers | 12 | 25 | 0.7 (0.3, 1.4) | 1.1 (0.5, 2.6) | 0.769 |
| | Others[€] | 28 | 78 | 0.5 (0.3, 0.9) | 0.6 (0.3, 1.1) | 0.131 |
| **Monthly income** | ≤ 28 Dollars | 112 | 141 | 2.9 (1.9, 4.3) | 2.73 (1.8, 4.2) | 0.0001 |
| | > 28 Dollars | 50 | 183 | 1.00 | 1.00 | |
| **Ever had chemical exposure** | No | 36 | 251 | 1.00 | 1.00 | |
| | Yes | 126 | 73 | 11.98 (7.62,18.85) | 11.9 (7.6,18.8) | 0.0001 |
| **Alcohol drinking** | No | 22 | 116 | 1.00 | 1.00 | |
| | Yes | 140 | 208 | 3.5 (2.1, 5.8) | 1.47 (0.82,2.65) | 0.198 |
| **Coffee drinking** | No | 22 | 83 | 1.00 | 1.00 | |
| | Yes | 140 | 241 | 2.2 (1.3,3.6) | 1.36 (0.75,2.48) | 0.316 |
| **Had a regular physical exercise** | Yes | 35 | 120 | 1.00 | 1.00 | |
| | No | 127 | 204 | 2.1 (1.3, 3.3) | 15.5 (5.7,42.3) | 0.0001 |
| **Types of chronic diseases** | Diabetes Mellitus | 25 | 73 | 1.00 | 1.00 | |
| | Liver disease | 4 | 32 | 1.1 (0.3, 4.7) | 0.3 (0.02,3.8) | 0.365 |
| | Hypertension | 3 | 53 | 0.2 (0.1,0.6) | 0.03 (0.005,0.2) | 0.0001 |
| | COPD | 35 | 82 | 1.1 (0.5,2.01) | 0.2 (0.06, 0.7) | 0.011 |
| | Others[#] | 20 | 84 | 0.4 (0.2,0.8) | 0.06 (0.01,0.2) | 0.0001 |
| **HIV status** | No | 109 | 189 | 2.2 (1.4,3.6) | 1.6 (0.5,4.8) | 0.399 |
| | Yes | 26 | 30 | 3.4 (1.7,6.6) | 1.2 (0.3,5.004) | 0.812 |
| | Unknown | 27 | 105 | 1.00 | 1.00 | |

COPD: chronic obstructive pulmonary disease; Others[#]: chronic kidney disease and cardiac failure

## Types of chronic disease

The current study revealed that the likelihood of risk for NHL among various chronic ill patients was different. Respondents who had hypertension, COPD, chronic kidney disease, and cardiac failure were less at risk for NHL compared with those who had diabetes mellitus. This study finding supported by a meta-analysis study [15] as DM patients were at high risk for NHL. However, the underlying mechanism is unclear. The possible justification for this result could be Diabetes Mellitus (type two) is an autoimmune disease, which may aggravate the chance of the NHL occurrence. Future studies should focus on elucidating potential pathophysiologic links between diabetes and NHL.

## Strength and limitation of the study

The current case-control study provides stronger evidence than a cross-sectional and descriptive studies. The current case-control study provides stronger evidence than a cross-sectional and descriptive studies. The selection of each case based on a histopathological confirmation

makes the study stronger. The limitations of this study could be the possibility of recall bias, the selection of controls, and the assessment of exposure and power issues.

Another limitation of this study was measurement error; though training on measurements and standard procedures given, it could not be a 100% perfect on the measurement of weight and height.

## Conclusions

This study identified the determinants of the NHL. Having a low monthly income, being male sex, ever had chemical exposure, not engaged in regular physical exercise, and being diabetic patient was at an increased risk for non-Hodgkin's lymphoma.

## Supporting information

**S1 File. English version questionnaire.**
(PDF)

**S2 File. SPSS data—with no identifiers.**
(SAV)

## Acknowledgments

We would like to thank data collators, supervisors and study patients for their contributions.

## Author Contributions

**Conceptualization:** Dessalegn Chekol, Melkamu Bedimo, Yihun Mulugeta, Getasew Mulat Bantie.

**Formal analysis:** Dessalegn Chekol, Getasew Mulat Bantie.

**Funding acquisition:** Dessalegn Chekol, Getasew Mulat Bantie.

**Investigation:** Getasew Mulat Bantie.

**Methodology:** Dessalegn Chekol, Melkamu Bedimo, Yihun Mulugeta, Getasew Mulat Bantie.

**Software:** Dessalegn Chekol, Melkamu Bedimo, Yihun Mulugeta, Getasew Mulat Bantie.

**Supervision:** Melkamu Bedimo, Yihun Mulugeta, Getasew Mulat Bantie.

**Validation:** Melkamu Bedimo, Getasew Mulat Bantie.

**Writing – original draft:** Dessalegn Chekol, Getasew Mulat Bantie.

**Writing – review & editing:** Melkamu Bedimo, Yihun Mulugeta, Getasew Mulat Bantie.

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
