## [Decision Letter · Decision Letter 0]

5 Aug 2020

PONE-D-20-14290

Determinants of Non-Hodgkin’s Lymphoma at Felegehiwot Specialized Hospital, North West Ethiopia : a case-control study

PLOS ONE

Dear Dr. Bantie,

Thank you for submitting your manuscript to PLOS ONE. After careful consideration, we feel that it has merit but does not fully meet PLOS ONE’s publication criteria as it currently stands. Therefore, we invite you to submit a revised version of the manuscript that addresses the points raised during the review process.

We look forward to receiving your revised manuscript.

Kind regards,

Yan Li

Academic Editor

PLOS ONE

Journal Requirements:

3. Please include additional information regarding the survey or questionnaire used in the study and ensure that you have provided sufficient details that others could replicate the analyses. For instance, if you developed a questionnaire as part of this study and it is not under a copyright more restrictive than CC-BY, please include a copy, in both the original language and English, as Supporting Information. In addition, please include further details of the pre-testing of this tool, including the number of participants and where they were recruited from.

4. You indicated that you had ethical approval for your study. In your Methods section, please ensure you have also confirmed that your IRB waived the need for parental consent for minors aged 13-17 and how parental permission was determined. In the manuscript you state: "13-17 years old patients,’ assents secured solely from them with parental/guardian permission".

5.We note that you have indicated that data from this study are available upon request. PLOS only allows data to be available upon request if there are legal or ethical restrictions on sharing data publicly. For information on unacceptable data access restrictions, please see http://journals.plos.org/plosone/s/data-availability#loc-unacceptable-data-access-restrictions.

6. Your ethics statement must appear in the Methods section of your manuscript. If your ethics statement is written in any section besides the Methods, please move it to the Methods section and delete it from any other section. Please also ensure that your ethics statement is included in your manuscript, as the ethics section of your online submission will not be published alongside your manuscript.

Additional Editor Comments (if provided):

editor comments

Please have this manuscript edited by some professional proofreading company.

Reviewers' comments:

Reviewer's Responses to Questions

**Comments to the Author**

1. Is the manuscript technically sound, and do the data support the conclusions?

Reviewer #1: Partly

2. Has the statistical analysis been performed appropriately and rigorously? 

Reviewer #1: I Don't Know

3. Have the authors made all data underlying the findings in their manuscript fully available?

Reviewer #1: Yes

4. Is the manuscript presented in an intelligible fashion and written in standard English?

Reviewer #1: No

5. Review Comments to the Author

Reviewer #1: The authors desired to study epidemiologic risk factors for NHL in their hospital in West Ethiopia. The most provocative finding is the association of increased risk with chemical exposure. This remains an important global health questions.

Abstract: Many places where parallel structure is needed to clean up the grammar. Many other grammatical errors throughout.

p.3 Background: Eliminate the first three paragraphs. Not all statements are accurate or relevant. Many are too rudimentary for a scientific paper in a major journal.

Consider condensing to a letter to the editor.

6. PLOS authors have the option to publish the peer review history of their article (what does this mean?). If published, this will include your full peer review and any attached files.

Reviewer #1: No

---

## [Author Response · Author response to Decision Letter 0]

5 Oct 2020

Dear reviewers and editors, good day to you all! We authors made extensive revisions and amendments as per your comments and guidance. We believe that the manuscript is easy to read and understand. The corrections are found in the clear and track changed manuscript. Thank you very much in advance.

---

## [Decision Letter · Decision Letter 1]

24 Nov 2020

Determinants of Non-Hodgkin’s Lymphoma at Felegehiwot Specialized Hospital, North West Ethiopia : a case-control study

PONE-D-20-14290R1

Dear Dr. Bantie,

We’re pleased to inform you that your manuscript has been judged scientifically suitable for publication and will be formally accepted for publication once it meets all outstanding technical requirements.

Kind regards,

Yan Li

Academic Editor

PLOS ONE

Additional Editor Comments (optional):

Reviewers' comments:

Reviewer's Responses to Questions

**Comments to the Author**

1. If the authors have adequately addressed your comments raised in a previous round of review and you feel that this manuscript is now acceptable for publication, you may indicate that here to bypass the “Comments to the Author” section, enter your conflict of interest statement in the “Confidential to Editor” section, and submit your "Accept" recommendation.

Reviewer #2: (No Response)

2. Is the manuscript technically sound, and do the data support the conclusions?

Reviewer #2: (No Response)

3. Has the statistical analysis been performed appropriately and rigorously? 

Reviewer #2: (No Response)

4. Have the authors made all data underlying the findings in their manuscript fully available?

Reviewer #2: (No Response)

5. Is the manuscript presented in an intelligible fashion and written in standard English?

Reviewer #2: (No Response)

6. Review Comments to the Author

Reviewer #2: (No Response)

7. PLOS authors have the option to publish the peer review history of their article (what does this mean?). If published, this will include your full peer review and any attached files.

Reviewer #2: No

---

## [Editor Report · Acceptance letter]

7 Dec 2020

PONE-D-20-14290R1 

*Determinants of Non-Hodgkin’s Lymphoma at Felegehiwot Specialized Hospital, North West Ethiopia: a case-control study*

Dear Dr. Bantie:

I'm pleased to inform you that your manuscript has been deemed suitable for publication in PLOS ONE. Congratulations! Your manuscript is now with our production department. 

Kind regards, 

on behalf of

Dr. Yan Li 

Academic Editor

PLOS ONE